# Review of Current Real-World Experience with Teriparatide as Treatment of Osteoporosis in Different Patient Groups

**DOI:** 10.3390/jcm10071403

**Published:** 2021-04-01

**Authors:** Barbara Hauser, Nerea Alonso, Philip L Riches

**Affiliations:** 1Rheumatic Disease Unit, Western General Hospital, NHS Lothian, Edinburgh EH4 2XU, UK; priches@exseed.ed.ac.uk; 2Rheumatology and Bone Disease Unit, Centre for Genomic and Experimental Medicine, MRC Institute of Genetics and Molecular Medicine, University of Edinburgh, Edinburgh EH4 2XU, UK; n.alonso@ed.ac.uk

**Keywords:** Teriparatide, anabolic treatment, osteoporosis, fracture

## Abstract

Teriparatide has proven effective in reducing both vertebral and non-vertebral fractures in clinical trials of post-menopausal and glucocorticoid-induced osteoporosis. Widespread adoption of Teriparatide over the last two decades means that there is now substantial experience of its use in routine clinical practice, which is summarized in this paper. Extensive real-world experience of Teriparatide in post-menopausal osteoporosis confirms the fracture and bone density benefits seen in clinical trials, with similar outcomes identified also in male and glucocorticoid-induced osteoporosis. Conversely, very limited experience has been reported in pre-menopausal osteoporosis or in the use of Teriparatide in combination with other therapies. Surveillance studies have identified no safety signals relating to the possible association of Teriparatide with osteosarcoma. We also review the evidence for predicting response to Teriparatide in order to inform the debate on where best to use Teriparatide in an increasingly crowded therapeutic landscape.

## 1. Introduction

It is now 20 years since the publication of the seminal clinical fracture prevention trial on Teriparatide PTH1-34 (TPTD) [1]. Further randomised controlled trials on TPTD [2,3,4] and more recently the head-to-head VERO trial with Risedronate [5] led to the approval and established use of TPTD as a first anabolic osteoporosis treatment.

TPTD is a recombinant human parathyroid hormone (1–34) that is given most commonly as a daily subcut injection. PTH, when given intermittently, stimulates bone turnover with increased osteoblast activity, leading to a steep rise in the bone formation marker P1NP and gain in bone density [6]. TPTD also stimulates bone resorption; however, the degree of bone formation over the initial 12 months outweighs the level of bone resorption, which leads to net bone formation leading to increased bone mass and bone strength, as shown in Figure 1 [1,2,7]. In the pivotal trial, patients gained 6% in spinal BMD over 12 months. In order to avoid bone loss after stopping TPTD treatment, it needs to be followed with antiresorptive treatment [7].

Bone formation is particularly pronounced in trabecular bone, which is reflected by a greater BMD increase at the spine than the hip. Initial trials have highlighted the TPTD efficacy in reducing the risk of vertebral fractures and the recent head-to-head trial with Risedronate confirmed that TPTD is superior to oral bisphosphonates in preventing vertebral fractures in patients with severe spinal osteoporosis [1,5]. However, to date, clinical trials failed to demonstrate superiority of TPTD in preventing non-vertebral or hip fractures when compared to the much cheaper alternatives such as oral or intravenous bisphosphonates [5,8].

In day-to-day practice TPTD is mostly prescribed to patients with severe osteoporosis who have sustained one or more vertebral fractures or non-vertebral fractures [9]. In the US a medical claims database analysis of 200 million patients showed that between May 2017 and September 2018 almost 10,000 patients were prescribed TPTD, which would equate to approximately 50 new TPTD patients per million per year [10], a number which is reflected by a single center cohort study in the UK which included 500 patients over 11.5 years in a health board of approximately 1 million patients [11].

At this stage, we have gained a wealth of clinical experience on the use of TPTD for severe osteoporosis mostly in postmenopausal women but also importantly in patient groups which fall outside the clinical trial population such as men with osteoporosis or patients with specific comorbidities [9,12,13]. In addition, clinical experience provides data on important aspects such as predictors to TPTD response and long-term TPTD effect. Observational data also helps answer questions which clinical trials have left open, such as the impact of TPTD on hip fracture risk, which a number of observational studies [14,15] and a meta-analysis [16] have recently tried to answer.

This narrative review will summarize the available real-world data on TPTD effectiveness and safety accrued from TPTD use in various patient groups including the less well studied groups such as pre-menopausal women, older patients and men with osteoporosis.

## 2. Teriparatide Data in Various Patient Groups

### 2.1. Teriparatide in Post-Menopausal Osteoporosis

In clinical trials of post-menopausal women TPTD has demonstrated effectiveness in improving bone mineral density (BMD) and reducing clinical fractures compared to placebo [1] and to oral bisphosphonates [17,18]. The VERO study remains the only comparative study concerning bisphosphonate with incident fracture as the primary outcome and confirmed that TPTD significantly reduced vertebral fracture risk in post-menopausal osteoporosis, though no significant reduction in non-vertebral fracture was seen [5]. A recent meta-analysis of clinical trials suggested a modest reduction in non-vertebral fractures is also seen with Teriparatide in comparison to bisphosphonate [19].

Real-world data has been gathered in large cohorts of patients treated with Teriparatide across the United States, Europe and Japan [9,20,21,22,23] and are detailed in Table 1. Outcome data is awaited concerning the ALAFOS cohort, which has recruited patients from Asia and South America [24] with such studies being important in establishing effectiveness across diverse genetic and environmental backgrounds. While no comparator population is available, an assessment of effectiveness has been made for 8878 patients pooled from these cohorts by comparing fracture outcomes in the first six months of treatment to the remaining treatment period (varying between 18 and 24 months). Female patients make up 92% of the group, with an average age of 71 years, very similar in age to the VERO trial cohort, which had a mean age of 72.6 years. This analysis has demonstrated a reduction in both vertebral and non-vertebral fractures in postmenopausal women, as well as a significant reduction in hip fractures for the cohort as a whole. Subgroup analysis in patients above or below age 75 showed significant reductions in clinical vertebral fracture and non-vertebral fracture in both groups; however, an interaction analysis suggested that the relative reduction in clinical vertebral fractures was smaller in the older subgroup at 49% compared to the younger cohort with 71% [9]. The authors speculate that this higher fracture rate in older patients concerning real-world experience could be driven by increased levels of comorbidities associated with fracture, which were excluded in previous controlled trials that did not show a connection to age [25]. Our group reported real world outcomes in post-menopausal women with severe osteoporosis treated with TPTD, in comparison to a standard care group unwilling or contraindicated from taking TPTD (details of the original and extended cohorts given in Table 1) [11,26]. Patients treated with standard care tended to be older (mean age of 74 years, compared to 69 years) than Teriparatide-treated individuals, and have more comorbidities, indicating that in routine practice older, more frail individuals might elect not to have TPTD therapy. Within this cohort TPTD was superior in preventing new vertebral fractures compared with those treated with standard care (4.8% vs. 10.1%) but there was no significant difference in the incidence of non-vertebral fractures [11]. Moreover, in keeping with data from randomised trials we found a significantly greater increase in BMD at the spine with TPTD, as compared with standard care, but no difference between groups in terms of the change in femoral neck BMD or total hip BMD. While the absolute change in bone density remained very similar in those patients followed up on for five years, the difference was no longer significant, presumably due to the observed greater variation in BMD and the much smaller number of patients concerning whom this data was available.

### 2.2. Teriparatide in Pre-Menopausal Osteoporosis

Treatment of pre-menopausal osteoporosis is not a recognised indication for Teriparatide, despite which there are numerous reports of outcomes in this population. TPTD has shown anabolic effects in a pilot study of idiopathic osteoporosis in 21 women with an average age of 39. An average increase of 9.8% in spine BMD was observed over 18 to 24 months, though four women (19%) were deemed non-responders to treatment [33]. While this is of interest given that bisphosphonate therapy should be avoided in women of childbearing potential, it is disappointing to see that a follow-up study from this cohort observed a 4.8% decline of BMD in the lumbar spine, albeit more modest decline was seen in the hip [34]. These changes led the authors to conclude that premenopausal women treated with TPTD should be considered for anti-resorptive therapy on completion of treatment, particularly those over the age of 40 in whom greater bone loss was observed. 

Case reports have described often dramatic improvements in bone density and back pain of pregnancy and lactation-associated osteoporosis following the use of TPTD. For example Lampropoulou-Adamidou et al. described 24.4% improvement in bone density and symptomatic improvement after 13 months [35], while Stumpf et al. describe 42.5% improvement after just six months of treatment [36]. Such reports are uncontrolled and do not take into account the expected recovery in bone density that is seen after weaning [37]. A real-world series of 27 cases of pregnancy and lactation-associated osteoporosis treated with 12 months TPTD has compared outcomes to five participants who declined therapy [38]. After adjustment for age and baseline bone mineral density the treated patients demonstrated a significantly greater gain in bone mineral density of 15.5% compared to 7.5% recovery of bone mineral density in the control population, with younger age associated with greater improvement in bone density. There is very limited long-term outcome data, or evidence comparing outcomes between active therapies with one retrospective case series with a follow up of 2.5 years reporting a mean gain of 14.9% bone density in the spine with TPTD, and 10.2% with bisphosphonate treatment, though realistically the numbers are too small to allow meaningful comparison to be made. In view of these limitations, concerns about safety and the longer term implications of treatment in young women then the use of Teriparatide in this context remain controversial, with these issues discussed in more detail in a recent review article [39].

### 2.3. Teriparatide in Men

Randomised controlled trial data demonstrate a lumbar spine BMD gain of 5.9% in men after 11 months (20 µg daily) [3] and 13.5% gain after 18 months (25 µg daily) [40], which is similar to that in women with 9.7% lumbar spine BMD gain after 12 months (20 µg daily) [1]. Randomised trial data on TPTD fracture risk reduction in men is scarce; however, a follow up analysis on 290 men who received 11 months of TPTD showed a trend of total vertebral fracture risk reduction and significant reduction of moderate to severe vertebral fractures in the TPTD group [2]. A randomized open label trial in GIOP in men has compared weekly Risedronate (35 mg) with daily Teriparatide (20 µg) treatment. After 18 months of treatment, TPTD results in a larger increase in spinal BMD and improved microstructure compared with Risedronate [41]. In many countries TPTD has not been approved for use for men and hence observational data is correspondingly limited [42]. In the large registry studies (JFOS, exFOS), approximately 10% of patients were men (see Table 1) and the number was generally too small to allow sub-analysis of fracture data. However, the pooled analysis of multiple registry studies led by Langdahl et al. [9] has shown that TPTD use in 710 men is associated with reduced clinical vertebral fractures from six months onwards compared to the reference period (0–6 months). The baseline non-vertebral fracture rate in men was low in the reference period, which might have been one of the reasons why the analysis did not show a TPTD period effect in this group. An important perceived difference between clinical trial populations and real-world studies seems to be the age of men who participated. The mean age in clinical trials ranges from 50 [40] to 59 years [3], which is substantially younger than the average age of subjects included in the real-world studies, whose ages range from 65 to 75 years, as outlined in Table 1. This issue may have an effect on study interpretation as we know that older patients are more likely to drop out of studies or die before study completion.

Severe osteoporosis is equally if not more devastating for men compared to women [43,44]. Over a third of men (37.1%) who sustain a hip fracture will die within the first year of fracture, which is a significantly higher rate than for women who will die in the same time frame (26.4%) [43].

The real-world experience of Teriparatide use in men does suggest reduction of vertebral fractures with the use of TPTD; however, more clinical trials and real-world studies are required to understand TPTD efficacy in men. Additionally, for future clinical trials it will be of utmost importance to ensure that the demographics of the clinical trial population will reflect the real-world population as accurately as possible concerning age.

There is undoubtedly a need for further studies on osteoporosis treatment in men with regular review of medicine licensing.

### 2.4. Teriparatide in Glucocorticoid Osteoporosis (GIOP)

Exogenous glucocorticoid administration is the most common cause of secondary osteoporosis. Fracture risk, in particular vertebral fracture risk, rises rapidly after the initiation of high dose glucocorticoids [45,46]. There is a pathophysiological rationale for using anabolic treatment in GIOP as glucocorticoid treatment decreases proliferation of osteoblast precursors and stimulates osteoblast and osteocyte apoptosis, which together leads to a rapid reduction of bone formation [47,48].

The first pivotal randomised controlled trial compared TPTD with daily Alendronate over 18 months in patients who had taken glucocorticoid treatment for at least three months [1,4]. TPTD treatment resulted in greater spinal and hip BMD gain and a significantly lower vertebral fracture rate than in the Alendronate group. There was no significant difference between both groups in non-vertebral fracture rates. Further clinical trial results showed that TPTD is superior to oral bisphosphonates in increasing lumbar spine BMD in GIOP in men and in post- and premenopausal women [41,49].

The analysis of a large medical and pharmacy claims US database showed that approximatly 12% of patients who received TPTD had a history of glucocorticoid long-term exposure which is double the percentage of patients prescribed Denosumab [10], indicating a prescriber’s preference for the use of TPTD in GIOP. Most multicentre observational studies have included patients who were concurrently treated with glucocorticoids; however, the frequency of GIOP patients varied from 2.7 to 14.4% (JFOS, ExFOS, EFOS), as outlined in Table 1. A pooled analysis of four observational studies which included 958 patients who have received glucocorticoids showed that GIOP patients have significantly higher fracture rates of all types compared to patients who have not received glucocorticoids. TPTD treatment reduced the incidence of clinical vertebral fractures by 73% and non-vertebral fractures by 24% after at least six months of treatment compared to the initial treatment period (0–6 months) in which TPTD had minimal effect [9]. A post-hoc analysis of the pivotal GIOP trial [4] suggests that high dose glucocorticoids (>15 mg/day) may attenuate the TPTD treatment effect [50]; however, to our knowledge this has not been confirmed in further clinical or observational settings. In view of the trial and real world outcomes, GIOP appears to be a particularly attractive target for TPTD. In current practice, TPTD is frequently prescribed for GIOP patients who suffer a fracture despite antiresorptive treatment [51]. However, it might be worth exploring early use of TPTD treatment in patients who are subjected to long term high dose glucocorticoids as a way of preventing irreversible bone loss and fractures.

## 3. Predictors for Teriparatide Response

One of the main limitations of TPTD treatment is variability in the patient response. The average increase in BMD after 18–24 months of treatment is 13%, but individual responses can range from no significant changes compared to BMD prior to treatment to a 53.4% increase in BMD. In about one fifth of patients the gain in BMD is less than 5%, a figure similar to the average increase after bisphosphonate therapy [26]. The cause of such variability in response to treatment is still unknown, and therefore it is difficult to predict which patients will benefit from TPTD treatment. Attempts have been made to identify markers of treatment response but with very limited success. In a study on 203 women treated with TPTD, Heaney and colleagues reported an inverse association between BMI and response to treatment at the lumbar spine BMD [52], but this finding could not be replicated [26]. P1NP biomarker has also been proposed as a possible predictor for TPTD response, since changes in serum levels at three months of TPTD treatment have been associated with BMD response at two years [53]. The difficulty with the use of P1NP change to determine therapy is that it may be more difficult to change treatment after a medication has been started and established.

In a retrospective review of almost 500 patients Elraiyah et al. [31] showed that prior bisphosphonate and Vitamin D treatment were associated with TPTD treatment failure, which was defined as spine BMD gain of <3% or sustaining one or more osteoporotic fractures after six months of treatment. These findings however have not been replicated in other observational studies and randomised controlled trials [5,54] have shown that TPTD response remains robust regardless of previous antiresorptive therapies.

It has also been hypothesized that the genetic background of the patient could determine the success of the TPTD treatment, as seen for other medications [55]. To date, there is only one genetic study in a small cohort of 42 postmenopausal women tested for the effect of Teriparatide on BMD and bone turnover markers according to the *VDR* BmsI genotype. This polymorphism within the vitamin D receptor gene showed an association with BMD and fracture rate in several studies [56,57,58,59]. However, no conclusive results were obtained regarding its involvement in Teriparatide response, probably due to the small sample size [60]. Further studies to screen the whole genome would be required to identify potential genetic markers that could predict the response to treatment and support the personalised management of these patients.

## 4. Evaluating the Risk of Osteosarcoma with Teriparatide

Pre-clinical studies identified a dose-dependent risk of osteosarcoma in rats treated with TPTD [61]. This finding was not replicated in subsequent primate or human studies; a black box warning of risk of osteosarcoma remains in place advising avoidance of Teriparatide in patients at risk of osteosarcoma, for example, due to previous radiotherapy to bone. Interim findings from a prospective registry study of patients treated with TPTD reported in 2018, by which stage 242,782 person years of follow up had been accrued and no incident osteosarcomas identified [62]. Whilst superficially reassuring, no conclusion can be drawn at this stage given the rarity of osteosarcoma, with the study authors noting that they anticipate being able to detect a fourfold increase in incidence if one exists by the close of the study, which would be the equivalent of one additional case per 123,000 person years. A related postmarketing surveillance study investigated 1173 patients with confirmed osteosarcoma and identified three cases with confirmed prior exposure to TPTD; this was considered to be no different than the expected background incidence of osteosarcoma, with the study being able to detect a three-fold increased incidence [63]. On an anecdotal level there is also reassurance from reports of successful outcomes with the use of TPTD in radiotherapy-induced osteonecrosis of the jaw [64].

## 5. Combination Treatment of TPTD with Antiresorptive Treatment

The combination of TPTD with antiresorptive treatment is not routinely used in clinical practice. In view of the fact that TPTD increases bone resorption and has a modest effect on non-vertebral fracture reduction, the combination of TPTD and anti-resorptive treatment was thought to be a promising strategy in order to combine the anti-resorptive and anabolic treatment effects. Initial trials with a combination of TPTD and oral bisphosphonates showed no clear benefit over TPTD alone [7,65]. On the contrary, the addition of Alendronate seemed to blunt the bone forming effect of TPTD [7]. Cosman et al. [66] showed that the combination of TPTD and Zoledronate led to a small but significant increase in total hip BMD over one year (2.3%) compared to TPTD treatment (1.1%) alone (*p* < 0.01). Recently, however, the DATA trials [67,68] showed that the combination of Denosumab and TPTD led to a substantially greater BMD gain at the spine and hip sites than one drug alone. The data is largely explained by the fact that Denosumab mitigates the TPTD-induced bone resorption increase. In order to explore how to maximise the TPTD effect, the DATA-HD trial [68] investigated whether the combination of high dose daily sc TPTD (40 µg) in combination with six monthly Denosumab injections is superior to the combination of standard TPTD (20 µg) dose and Denosumab. After 15 months of treatment the spine and hip BMD increase (17.5% and 6.8%, respectively) in the high dose TPTD group was significantly greater than with the standard TPTD treatment dose. A rapid, substantial BMD increase may be of particular interest for patients who sustained multiple recent fractures and who are at very high risk of sustaining a further fracture [69]. The rationale for rapid BMD increase is the imminent risk, which means that patients are at the highest risk in the first two years after they have sustained the fracture [70]. Additionally, the combination treatment may be of interest for patients who have severe osteoporosis at both the spine and hip levels, as the above DATA studies demonstrate significant gains at both sides.

However, none of the above-mentioned studies were powered to investigate the effect on fracture risk and the current evidence does not support combination therapy in routine use. Hence, currently there is little real-world data on combination therapy available, but some special cases, particularly those at very high fracture risk, may qualify for combination therapy in future and it will be important to collect observational and long-term data on patients who have been treated with this approach.

## 6. Conclusions

Current real world experience confirms that TPTD treatment effectively reduces the risk of vertebral and non-vertebral fractures in postmenopausal women and in men, and concerning glucocorticoid-induced osteoporosis [9]. Observational data also confirms that TPTD effectively reduces the risk of hip fractures by about 50% [9,15]. Finally clinical trial data has been replicated in real-world experience, which confirms the superiority of TPTD to oral bisphosphonates in vertebral fracture reduction in post-menopausal osteoporosis; however, the jury is out on whether TPTD is superior to anti-resorptive medication in non-vertebral fracture risk reduction. This experience supports a move to first-line use of Teriparatide in patients at high risk of vertebral fractures, with strong evidence to justify this approach in post-menopausal osteoporosis.

Looking at various patient groups, TPTD may be more effective in younger patients compared to patients over 75 [9]; however, this finding may be confounded by case selection and the fact that older patients have more comorbidities which can influence BMD and fracture risk. Real-world data on the use of TPTD in pre-menopausal women is restricted to BMD data only [33,34], which makes further collection of clinical experience in these patient groups even more important. Similarly, combination treatment, although not licensed, may be used more commonly in clinical practice for difficult cases in future and the collection of observational data will help to understand better whether and when to consider a combination approach. Given the limited experience in premenopausal women and for combined TPTD therapy, no recommendation for its use in these settings can be made.

Lastly, postmarketing surveillance and observational data are vital tools to evaluate long-term safety of new medications and provide invaluable reassurance when counselling patients for new treatments. Despite initial safety signals in rodent studies, long-term data on TPTD shows a reassuring safety profile [62,63].

## 7. Future Direction

The latest European International Osteoporosis Foundation (IOF) and European Society for Clinical and Economic Evaluation of Osteoporosis and Osteoarthritis (ESCEO) guidelines recommend the use of anabolic treatment first line for patients with very high fracture risk [69]. In addition, the approval and clinical use of TPTD biosimilars in many countries strengthens the cost-effectiveness arguments for anabolic treatments [71,72]. The new guidelines and cheaper medication cost may well increase TPTD use in daily clinical practice in the near future. Additionally, the anabolic repertoire over the last few years has been widened by the approval of Abaloparatide in the US in 2017 [73] and more recently by the approval of the monoclonal antibody to sclerostin (Romosozumab) in the US, Canada, Europe and Japan [74]. Both drugs have been licensed for use in postmenopausal women at very high fracture risk and hence will be in competition with Teriparatide [74,75,76]. Real-world studies will give additional insight about these new drugs and, in the absence of anabolic head-to-head trials, observational studies may allow cautious comparisons of efficacies, safety profiles, persistence and adherence to the current available anabolic treatment options. The therapeutic landscape for osteoporosis will undoubtedly be influenced by cost–benefit calculations; however, further studies into predictors of treatment response and real-life effectiveness may allow better targeting of patients at high risk and maximising of treatment effects.

## Figures and Tables

**Figure 1 jcm-10-01403-f001:**
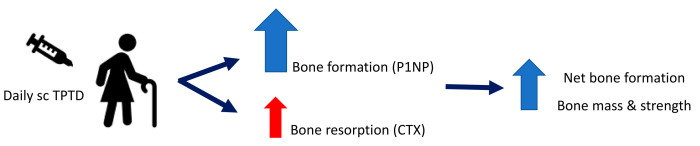
Mechanism of action of subcutaneous (sc) Teriparatide (TPTD). Teriparatide stimulates bone formation, reflected by an increase in the bone formation marker P1NP (procollagen type 1N propeptide) and increases bone resorption, as reflected by an increase in the bone resorption marker CTX (c-terminal telopeptide). Bone formation is greater than bone resorption leading to net bone formation with increased bone mass and strength.

**Table 1 jcm-10-01403-t001:** Real world observational studies of Teriparatide.

First Author	Year	Journal	Type of Study	Cohort	Patients on TPTD	Mean Age	Assessment Time (Months)	Post-Menopausal	Premenopausal	GIOP	Men	Ethinicity
Langdahl [21]; Ljunggren [27]	2016/2014	CTI/CMRO	MO	ExFOS	1454	70.3 ± 9.8	24	✔	✔ (1.8%)	✔ (14.4%)	✔ (9.4%)	Caucasians
Napoli, N [9]	2018	Bone	MO	Extended ExFOS	998	(as ExFOS)	42	✔	✔ (1.5%)	✔ (14.6%)	✔ (9.3%)	Caucasians
Soen [22]; Soen [28]	2017/2015	CMRO/CMRO	Observ.	JFOS	1996	76.9 ± 7.9	24	✔	✔ (12.1%)	✔ (2.7%)	✔ (9.9%)	Japanese
Rajzbaum [29]; Jakob [30]	2008/2012	CMRO/EJE	MO	EFOS	1645	71.5 ± 8.4	36	✔	x	✔ (14.4%)	x	Caucasians (86.8%)
Langdahl [9]	2018	Bone	4 MO	EFOS, ExFOS, JFOS, DANCE	8828	70.9 ± 10.6	various	✔	as outlined in single studies	✔ (10.9%)	✔ (8.1%)	Caucasians (ca 90%)
Silverman [23]	2013	OI	MO	DANCE	3720	68.0 ± 11.7 65.1 + 13.1 (men)	24	✔	unknown	unknown	✔ (9.9%)	Caucasians (88.2%), Hispanic (8.6%), African (1.5%)
Oswald [26]	2014	CTI	Observ.	single-centre cohort	217	69.8 ± 9.1	66	✔	x	✔ (4.1%)	✔ (6.9%)	Caucasians (predominantly)
Oswald [11]	2019	CTI	Observ.	single-centre cohort	496	69.6 ± 9.9	138	✔	x	✔ (6.9%)	x	Caucasians (predominantly)
Elraiyah [31]	2016	Bone Rep	Observ.	single-centre cohort	494	66.2 ± 12.6	14	✔	x	✔ (21.7%)	x	Caucasians (95.6%), African American, Asian
Chen [32]	2020	Clin Interv Aging	MO	ALAFOS	1136	75 ± 9.6	x	✔	x	✔ (4.0%)	x	East Asia (Taiwan, South Korea, China, Hong Kong)

✔ = included; x = not included; CTI = Calcified Tissue International; CMRO = Current Medical Research and Opinion; EJE = European Journal of Emdocrinology; OI = Osteoporosis International; Bone Rep = Bone Reports; Clin Interv Agin = Clinical Interventions in Aging; MO = Multinational Observational; Observ = Observational.

## Data Availability

Not applicable.

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
