# Peer review of "Review of Current Real-World Experience with Teriparatide as Treatment of Osteoporosis in Different Patient Groups"

_jcm, 2021, doi:10.3390/jcm10071403_

Round 1

Reviewer 1 Report

Dear Authors,

I read with interest your manuscript and I think that it is good quality paper suitable for publication.

Osteoporosis is defined as “a systemic skeletal disease characterized by low bone mass and microarchitectural deterioration of bone tissue with a consequent increase in bone fragility and susceptibility to fracture”. Approximately 40–50% of women sustain osteoporotic fractures in their lifetime. Osteoporosis is most prevalent in women over the age of 50 as the hormonal influence of estrogen on bone health dissipates with the onset of menopause. The progressive changes in bone structure, quality and density lead to pathological fractures and an increase in morbidity and mortality among menopausal women. Osteoporosis and consequent fracture are not only limited to postmenopausal women. There is increasing attention being paid to osteoporosis in older men. Men at high risk for fracture include those men who have already had a fragility fracture, men on oral glucocorticoids or those men being treated for prostate cancer with androgen deprivation therapy. Beyond these high risk men, there are many other risk factors and secondary causes of osteoporosis in men.  Despite some recent increased attention in men, osteoporosis is still considered a disorder of postmenopausal women.

In my opinion the submitted paper is very valuable because the authors summarized the available data on the efficacy and safety of TPTD in different patient groups, including less studied groups such as premenopausal women, older patients and men with osteoporosis. Therefore, I recommend this article for publication.

Reviewer 2 Report

The paper entitled "Review of current real-world experience with Teriparatide" by Hauser et al. summarizes studies that have examined the use of Teriparatide in clinical settings, deemed "real-world" by the authors, as opposed to controlled clinical trials. Overall, the paper is interesting and contributes to the field in its summary. However, additional points are needed: 

  1. The introduction should give more background information on Teriparatide. What is the target clinical population for this drug (indications for use)? What is the mechanism of action (basic biology - PTH signaling and bone turnover)? How often is it prescribed in a clinical setting?
  2. The section on Teriparatide use in pre-menopausal osteoporosis is very limited. I understand the difficulty because of the relative lack of real-world data, but I'm not sure a single paragraph justifies it being in this review. Please expand. 
  3. Please address the ages of the men in the studies in the "Teriparatide in Men" section of the manuscript. How does age at time of TPTD prescription affect real-world data collection? 
  4. Expand on the statement "we know that severe osteoporosis is equally if not more devastating for men than women". What are the data to back this statement up? 
  5. Please define what the checkmarks versus Xs versus ?s in the table mean. What does 323 (217 cases) mean for example in the Oswald line? What does "as above" mean in line 5? Tables should be able to stand on their own, these needs to be cleared up. 
  6. In line 154, are you using "anabolic therapy" as an equivalent for TPTD? What about Abaloparatide? Be specific in your language. (Same goes for line 156). 
  7. Lines 182-184 are unclear in how "SNPs on the VDR gene" relate to TPTD. 
  8. Break up the run-on sentence in lines 189-192. 
  9. The statement "Teriparatide increases bone resorption" in line 205 is going to be confusing to readers unfamiliar with the drug if you do not explain the mechanism of action in the introduction.

Reviewer 3 Report

This review focused on Teriparatide treatment in various patient groups, which is fluently written and is beneficial for clinical practice. The main idea of the review needs to be further highlighted and the legibility needs to be enhanced, maybe more illustrated graphs and tables could help. The detailed recommendations are as follow:

  1. The title of this review is somehow broad which is suggested to be more focused on the topic.
  2. In the Introduction section, the importance and significance of this review should be emphasized. For example, how is the current relationship between Teriparatide and osteoporosis treatments? Why is Teriparatide of concerned rather than other anti-osteoporotic drugs? Are there any controversial points regarding the use of teriparatide?
  3. It is creative to explore the relationship between Teriparatide and different patient groups, which is the most important part of this review. The author summarized the real-world data of Teriparatide well, however an overall summary at the end of this part is expected to show which group is more favor Teriparatide treatments and its potential mechanism.
  4. Abbreviations in Table 1 should be well defined. Moreover, please check if the table include all qualified studies? (eg. PMCID: PMC6310708; PMCID: PMC4457272)
  5. Other adverse events should be mentioned in “Safety data in the long-term use of teriparatide”, or change the title of this part if the author just wants to illustrate the relationship between Teriparatide and osteosarcoma.
  6. Combination treatment of Teriparatide is also of great interest for clinical recommendations. Actually, in addition to anti-resorptive drugs, the combined Teriparatide and Denosumab treatment has shown advantages to increase BMD for example. Maybe more information could be added in this section. Also,
  7. Combination treatment is always based on pharmacological properties and biological mechanisms. Analysis or speculation from this perspective after presenting real-world data, could better illuminate the potential causes of clinical outcomes and the direction of future research.

Round 2

Reviewer 2 Report

Thank you to the authors for their review work on this paper. 

Reviewer 3 Report

All comments are responded, and related changes have been made in the updated version of manuscript.